# Menstrual Symptoms: Insights from Mobile Menstrual Tracking Applications for English and Chinese Teenagers

Sisi Peng [1,*], Yuyin Yang [2] and Martie G. Haselton [1]

1   Department of Communication, University of California, Los Angeles, CA 90095, USA; haselton@ucla.edu
2   Department of Social Anthropology, University of Cambridge, Cambridge CB2 3RF, UK; yy530@cam.ac.uk
*   Correspondence: s.peng@ucla.edu

**Abstract:** Mobile software applications (apps) have transformed how individuals oversee and maintain their own health. One way that girls can monitor their menstrual cycles is through the increasingly widespread use of mobile menstrual tracking apps. This study aimed to examine menstrual symptom tracking for adolescents in English and Chinese apps, exploring menstrual literacy, cross-cultural differences, and framing, or presentation, of symptoms. The mixed-methods content analysis involved 15 popular free menstrual tracking apps in English ($n = 8$) and Chinese ($n = 7$), sampled from December 2022 to January 2023. A quantitative analysis of qualitative data was conducted through manual coding of content and automatically analyzing sentiment, or emotional tone, using a computational approach. We found that (1) menstrual literacy on symptom management or treatment was generally insufficient, (2) there were more available emotional than physical symptoms in English than Chinese apps, and (3) symptoms were framed more negatively than positively somewhat more in Chinese than English apps. Our findings emphasize the urgency to provide better evidence-informed communication about symptoms, either presented more positively or neutrally, in menstrual tracking apps for adolescent users. Since adolescence is a critical developmental stage that requires ample support, we recommend that digital menstrual trackers be crucially improved and future research should investigate how they can uniquely shape attitudes and experiences, and subsequent sexual and reproductive health empowerment and bodily autonomy.

**Keywords:** adolescents; menstruation; sexual and reproductive health; informatics; mobile health

## 1. Introduction

Growing up in the digital age, more and more adolescents are familiar with information and communication technologies. In 14 education systems across the Global North (e.g., U.S., Germany) and Global South (e.g., Chile, Uruguay), 94% of eighth grade students on average have access to the Internet at home [1]. In the U.S., 95% of adolescents had access to a smartphone in 2022, compared to 73% in 2014 [2]. In China, over 90% of adolescents under the age of 18 years accessed the Internet through smartphones in 2018 [3]. It is unsurprising that digital technologies are becoming more prevalent and have been assuming increasingly important roles in healthcare and health communication [4–6]. From a communication studies perspective, mobile software applications (apps) are personal informatics, as they are designed to help people gain self-knowledge (i.e., one's "behaviors, habits, and thoughts") and reflect on it [7]. Personal informatics is central to digital health and risk communication [8,9]. From a medical anthropology point of view, which underscores that health and illness are shaped by cultural, historical, political, and economic contexts, health tracking apps have been discussed in relation to neoliberalism or capitalist policies that foster markets not regulated by the government [10]. Under neoliberal practices, digital health technologies direct health management away from the healthcare system and toward the individual by allowing self-monitoring and urging self-responsibility [11,12]. For example, fitness trackers permit users to closely watch over their own health using a

wearable device rather than relying on frequent visits to healthcare providers. Advocates support the potential of this voluntary self-tracking to promote patient engagement, self-care, and preventive health behaviors, whereas critics argue that self-monitoring could bring about over-medicalization of the self and diminish the influence of structural barriers to care [11,12]. There are benefits to direct access to health information; however, from a public health standpoint, some major concerns with mobile health (mHealth), or the use of digital technologies for public health and clinical practice, are the sources and accuracy of information within a predominantly unregulated environment [13].

## 1.1. Menstrual Tracking

Hundreds of menstrual tracking apps exist, with estimated combined downloads of 500 million worldwide [14]. Most of them allow users to document cycle characteristics, such as the length, flow, symptoms, and sexual behavior. While many apps self-claim to enhance education around menstruation, a study of 17 menstrual tracking apps revealed that most apps are unsatisfactory in achieving this goal [15]. In particular, menstrual literacy, which is defined as "a baseline of knowledge and skills with regard to understanding the anatomical and biological facts of menstruation, caring for the menstruating body, and carrying out tasks related to menstrual care", was lacking [15]. For example, most apps allowed users to track symptoms, yet none provided information for teenagers to self-manage or treat symptoms [15]. In another study that examined popular menstrual tracking apps from a human–computer interaction lens using a mixed-methods approach, it was found that women's needs were not adequately met [9]. Participants perceived the apps to be unreliable in predicting periods and biased against sexual and gender minorities [9]. Most apps also fail to take into consideration the life-stage development of users, with minimal differentiation for girls who recently experienced menarche, or the first period, women who have been menstruating, or women who are approaching menopause [9,15]. Only a few apps that are designed to prevent unintended pregnancies have been tested with clinical trials [16], with the rest ineffective in serving contraceptive purposes [15,17].

Aside from practical purposes, many menstrual tracking apps promote the message of empowerment and advertise their products as means of "understanding one's body better, avoiding stain stigma, or as natural birth control" [18]. However, closer examinations of these messages reveal that menstrual tracking apps might, in reality, be perpetuating menstrual stigma by suggesting the maintaining of dignity through secrecy and further medicalizing normal bodies as needing to be fixed. While menstrual tracking apps have the immense potential to alleviate health disparities and empower girls to understand their cycles, girls may be exposed to insufficient menstrual literacy and period negativity as the norm at an early age.

## 1.2. Menstrual Stigma

Menstruation has a long history of being taboo, resulting in negativity and confusion toward a normal bodily process. Enduring cultural and religious beliefs and practices consider menstrual blood as unclean or impure [19]. In an empirical study, known as the "tampon experiment", researchers revealed that participants who saw a female confederate drop a tampon, versus a hair clip, out of her handbag led to more unfavorable judgments of her—and general objectification of women [20]. These findings demonstrate that menstruating women are, on average, deemed as less competent and likable than non-menstruating women. In part due to feelings of disgust toward menstrual blood and signs or reminders of periods, like the presence of menstrual products, being tied to negative perceptions of women who menstruate. Existing stereotypes of menstruating girls and women as "ill, disabled, out-of-control, unfeminine, or even crazy" further contribute to menstrual stigma [19].

Here, we begin our cross-cultural research by comparing and contrasting menstrual tracking technology in English and Chinese. Previous research highlighted that people's attitudes, interpretations, and experiences of menstruation are subject to socio-cultural

norms [21], and it is important to study menstruation in various contexts. In particular, English and Chinese are the two most widely spoken languages in the world, with about 1.5 billion and 1.1 billion people using them as their native or second language, respectively [22]. For example, in the U.S., the country with the highest total number of English speakers, there have been ongoing efforts to address misconceptions about menstruation and increase activism in promoting period positivity. However, there are mixed messages about menstruation that it is a normal bodily process but should be hidden [23]. Compared to the U.S., China is more socially conservative, and menstruation is treated with more negativity [23]. Menstrual blood is viewed as the worst bodily excretion and is associated with danger, pain, and death [23]. These similar but different perspectives on menstruation prompt the juxtaposition. Such social norms may be expressed in the prevailing narrative around periods in popular English and Chinese digital menstrual trackers.

Menstrual symptoms can also vary by culture. For instance, a cross-cultural study on premenstrual distress uncovered that Chinese women in Hong Kong reported fewer emotional and mental well-being symptoms, such as pain, concentration, and affect, than Caucasian and African American women in the U.S. [24]. However, there is also a widely held perception that Chinese people are not as emotionally inclined as English counterparts even though Chinese women still report emotional menstrual symptoms [25]. It is possible that differences in cultural perception and social norms toward menstruation influence people's understanding, bodily sensation, and exhibition of menstruation-related discomfort. These differences, in turn, could be displayed through the default options for symptoms provided by apps in English and Chinese.

### 1.3. Study Objectives

This research focuses on English and Chinese menstrual tracking apps for adolescents and investigates the symptoms provided for users. This study critically analyzes how menstrual symptoms reflect socio-cultural norms in such contexts. Particularly, we concentrate on what and how menstrual symptoms are communicated in popular free digital menstrual trackers as well as where there are information gaps for adolescent audiences, calling attention to how such apps can be beneficial or harmful for girls' sexual and reproductive health as well as opportunities for this technology to be improved.

### 1.4. Study Hypotheses

The goal of this study was to uncover the availability of menstrual literacy, any cross-cultural differences in menstrual symptoms, and the framing, or presentation, of menstrual symptoms in apps for adolescents.

Based on prior research, we predicted that there would be insufficient menstrual literacy on symptom management or treatment across apps. Menstrual literacy was operationalized as the number of resources for users to seek help and support for menstrual symptoms. We expected that Chinese apps would have less emotional symptoms compared to physical symptoms for users to track than English apps. We also expected that the symptoms in Chinese apps, relative to English apps, would be framed more negatively, which was operationalized as the sentiment, or emotional tone, of the content.

## 2. Materials and Methods

### 2.1. Study Design

A mixed-methods content analysis, which involved manual (human-assisted) and automated (computer-assisted) quantification of qualitative data, Ref. [26], was conducted of the top 8 English and 7 Chinese free apps (*N* = 15) from the Apple App Store. The reviewed apps were Clue, Flo, Eve, MagicGirl, My Calendar, Spot On, Period Tracker Lite, Easy Period Lite, Meet You, Period Helper, Soft Soft, Lemon Piggy, Grapefruit Menstruation, Grapefruit Auntie, and Easy Period. Most of the English apps assessed were developed in the U.S.; however, such apps are used by English speakers around the world. The total sample size was based on a previous study that investigated menstrual tracking apps,

which found that a purposive sample of at least 15 apps was adequate [15]. The App Store was searched using variations of relevant keywords involving adolescents or teenagers and period tracking. All apps were publicly available and purposefully chosen based on their age ratings (10–18 years old), higher average user ratings (scale of one to five stars), and larger total number of ratings (higher sum representing greater popularity).

### 2.2. Data Collection

Data were sampled and collected between December 2022 and January 2023. The authors downloaded the apps ($N = 15$) and inspected their features. All available symptoms were systematically logged on a shared spreadsheet. The options in Chinese apps were translated from Chinese to English by two independent coders fluent in both languages, which received near perfect agreement (Cohen's kappa = 0.896).

### 2.3. Data Analysis

There were three parts to this mixed-methods approach to the content analysis. The first part involved observing whether the menstrual trackers provided the ability to document symptoms and availability of resources for managing or treating symptoms as a way of assessing menstrual literacy. The second part consisted of developing a comprehensive codebook with categories for the tracking options, then manually coding the collected qualitative data. Based on the U.S. Department of Health & Human Services definitions of menstrual symptoms [27] and premenstrual syndrome (PMS) [28], physical and emotional are the two types of symptoms. In the digital menstrual trackers, there were a range of other health and well-being factors to monitor. Other categories that users were able to track were fertility (i.e., signs of ovulation, types of contraception), lifestyle (i.e., factors such as diet, exercise, sleep), medical (i.e., abnormal health issues), menstrual (i.e., cycle characteristics like flow and blood color), and sexual (i.e., sexual desire and activity). Two independent human coders were trained to analyze the categories. The interrater reliability for the English apps (Cohen's kappa = 0.941) and Chinese apps (Cohen's kappa = 0.939) reached almost perfect agreement. We also explored any cultural differences for English and Chinese audiences. The third part entailed a sentiment analysis of physical and emotional symptoms using a dictionary-based computational technique to automatically detect the tone communicated in text. A validated word–emotion association dictionary from the National Research Council Canada [29] was applied to the symptom dataset. Symptoms were broken down by words and each word was assigned a sentiment value based on the pre-annotated negative or positive sentiment words in the dictionary. Analyses were performed in Google Sheets and R version 4.2.3.

## 3. Results

### 3.1. Menstrual Literacy

All the apps ($N = 15$) served to log menstrual cycle information on a calendar and provided options for users to track symptoms. However, they did not all offer educational resources for users to manage or treat symptoms. In the English app sample ($n = 8$), five (63%) provided resources and three (37%) did not. For instance, there were articles and videos on how to relieve menstrual cramps, but emotional changes were not addressed. A few apps provided a chat or forum feature that facilitated peer-to-peer conversations among teenage users. Only one of the apps connected users directly to chat with a health educator. In the Chinese app sample ($n = 7$), three (43%) provided resources and four (57%) did not. For example, there were articles that gave diet recommendations and advice on menstrual pain. Similar to the English apps, the Chinese apps did not discuss how to address emotional symptoms. One of the apps allowed users to communicate with a healthcare provider and another app had a feature for users to make friends with peers.

*3.2. Symptom Type*

There were two types of menstrual symptoms: physical and emotional, which are broken down in Table 1. In the English app sample (*n* = 8), there were 150 (60%) emotional and 101 (40%) physical symptoms. The top three emotional symptoms were happy, sad, and anxious. The top three physical symptoms were cramps, acne, and tender breasts. In the Chinese app sample (*n* = 7), there were 52 (34%) emotional and 101 (66%) physical symptoms. The top three emotional symptoms were sad, unhappy, and very happy. The top three physical symptoms were dizziness, headache, and diarrhea.

**Table 1.** Counts of Categories among English and Chinese Menstrual Tracking Apps.

| Categories | English Apps (*n* = 8) | Chinese Apps (*n* = 7) |
|---|---|---|
| Symptoms | | |
| Emotional | 150 | 52 |
| Physical | 101 | 101 |
| Total | 251 | 153 |
| Fertility | 45 | 23 |
| Lifestyle | 32 | 25 |
| Menstrual | 30 | 32 |
| Sexual | 25 | 7 |
| Medical | 11 | 32 |
| Total | 143 | 119 |

Beyond symptoms, there were five other categories: fertility, lifestyle, medical, menstrual, and sexual. Other options that users can track by category are shown in Table 1. In the English app sample (*n* = 8), the top three other categories were fertility (45/143, 31%), lifestyle (32/143, 22%), and menstrual (30/143, 21%). In the Chinese app sample (*n* = 7), the top three other categories were menstrual (32/119, 27%), medical (32/119, 27%), and lifestyle (25/119, 21%). In the sexual category, there were 25 (17%) options to track in the English sample and 7 (6%) in the Chinese sample. Sexual items in the English apps consisted of protected/unprotected sex, masturbation, and feeling horny. Sexual items in the Chinese apps were worded as love making and there was no tracking of sexual desire.

*3.3. Symptom Framing*

Physical and emotional symptoms, broken down by word units, were further investigated. In both samples, there was more negative than positive sentiment, exhibited in Table 2. In the English app sample (*n* = 8), 87 (66%) symptoms were negative and 44 (34%) were positive in tone. The top three negative words were pain, anxious, and angry. The top three positive words were happy, tender (breast), and love. In the Chinese app sample (*n* = 7), 93 (76%) words were negative and 30 (24%) were positive in tone. The top three negative words were pain, dizziness, and headache. The top three positive words were happy, tenderness (breast), and moderate (pain). Limitations of classifying single words are addressed in the Discussion section.

**Table 2.** Counts of Negative and Positive Sentiments among Physical and Emotional Symptoms in English and Chinese Menstrual Tracking Apps.

| Category | English Apps (*n* = 8) | Chinese Apps (*n* = 7) |
|---|---|---|
| Negative | 87 | 93 |
| Positive | 44 | 30 |
| Total | 131 | 123 |

## 4. Discussion

Our study explored menstrual tracking apps for English and Chinese teenagers, specifically concentrating on symptoms. These apps aim to support users in tracking their menstrual cycle and symptoms; however, menstrual literacy on symptom management or treatment was generally inadequate. From a health communication perspective, information availability and sources are important for promoting healthy behaviors. In this mixed-methods content analysis of 15 apps, symptom tracking was available to users, but educational resources to support symptoms and find help can be improved. The majority of the apps lack the ability to contact a healthcare provider, which is a protective factor that reduces negative health outcomes and an essential feature that bolsters health and well-being. This finding is consistent with past research within health informatics [15]. Credible sources of information that cite scientific literature and involve healthcare professionals are also crucial for supporting adolescents who can be particularly susceptible to social influence. Social support for adolescents is pivotal in addressing health concerns as well. A minority of the analyzed apps connected adolescent users with other teenagers, indicating that digital health tools can be beneficial for developing peer relationships and building a sense of community among girls who are newly menstruating. Peer connection and support can help with navigating this sensitive period of life.

When assessing the type of symptoms, more emotional symptoms were available for English users to track than Chinese users. On the other hand, Chinese apps offered more options for physical, or somatic discomfort, than English ones. This cross-cultural difference in features between English and Chinese apps could be interpreted in a few ways. Firstly, this could originate from the pathologizing of women's emotions in Western culture. For instance, Ussher and Perz argued that American women diligently track their periods with extra caution against any signs of negative emotions and subsequently engage in self-monitoring, self-policing, and self-silencing [30]. Engaging in self-care practices, like taking a break from household tasks, might be considered a remedy for PMS, whereas normal emotional reactions to stressful events, such as irritability and moodiness, might be dismissed and explained away as "symptoms" [31]. Secondly, this finding aligns with previous work that pointed out that there is a misconception that Chinese women do not experience emotional menstrual symptoms [25]. In Eastern culture, echoing the classical studies in medical anthropology that focused on the presentation of depression in the Chinese population, Chinese people tend to report more somatic complaints in place of psychological ones [32]. Fear of stigma against mental illnesses and a cultural discouragement of expressing negative emotions or relationship strain have contributed to the Chinese population's skewed presentation of physical discomforts. Therefore, it is possible that menstruation-related symptoms were also somaticized in apps and are illustrated through the availability for reporting more physical rather than emotional distress. Interestingly, there were more options for tracking sexual activities available in English apps than in Chinese apps. Chinese apps also used love making as a euphemism for sexual intercourse, reflecting a more conservative, and potentially more stigmatized, social norm around sex among teenagers in China [33].

Symptoms were framed more negatively than positively, consistent with prior work on menstrual stigma [18], within English and Chinese apps, respectively. When comparing negative sentiments, Chinese apps had slightly more than English apps. Additional research is needed to understand the accuracy of information as well as the underlying reasons behind these differences in default symptom presentations in English and Chinese apps. Nevertheless, this study suggests that there may be possible priming effects in menstrual tracking apps. Past research has revealed that people's menstrual experiences are highly subject to priming. For example, previous knowledge of PMS could lead to an increase in reporting of negative, PMS-related symptoms [34]. In addition, participants who filled out the Menstrual Joy Questionnaire in the first week and the Menstrual Distress Questionnaire in the second week scored significantly higher on the "menstruation as a natural event" subscale (e.g., "menstruation is a reoccurring affirmation of womanhood",

"menstruation allows women to be more aware of their bodies") of the Menstrual Attitudes Questionnaire than their counterparts who filled out the two questionnaires in the reverse order [35,36]. This showed that participants can be primed to report more positive experiences of menstruation. Therefore, while menstrual tracking apps provide convenience and strengthen self-knowledge, it is also important to keep in mind that more negative framing of menstruation can prime more negative attitudes and experiences as well as further pathologize menstruation at an early age during adolescence.

### 4.1. Implications

Despite the lack of menstrual literacy and mostly negative framing of menstrual symptoms in the popular free apps assessed in this study, digital trackers could be advantageous in empowering girls with information about their own bodies. Consequently, self-tracking of menstrual cycles using accessible digital health tools could provide girls with more personal awareness and control over decisions related to their own sexual and reproductive health, facilitating greater bodily autonomy. Given that adolescents are constantly connected online, there are still opportunities for menstrual tracking apps to uniquely meet teenagers where they are. Digital menstrual trackers can help adolescents remember their last period and predict their upcoming period. Information from apps could be used to inform discussions between teenagers and healthcare providers, parents, or caregivers. Additionally, it is important that healthcare providers, parents, and caregivers are well informed about the advantages and disadvantages of menstrual tracking apps, so they can be a part of the adolescent user's self-tracking journey if chosen to do so. There should also be more menstrual tracking apps specifically tailored to teenage audiences, since the majority of the apps analyzed here appear to be geared toward adult populations. Areas of improvement in menstrual tracking apps for adolescents include increasing evidence-based information on symptom management and treatment that is easy to understand for young girls, enhancing involvement of healthcare professionals, cultivating peer relationships, and presenting positive, or neutral, information in order to encourage corresponding attitudes toward menstruation, as well as aiding conversations between parents and teenagers. Last but not least, policy makers need to ensure that there are data privacy and confidentiality protections in place for digital menstrual tracking users.

### 4.2. Strengths, Limitations, and Future Directions

To our knowledge, this is the first study that probes mobile menstrual tracking apps in both English and Chinese. Comparing and examining such technology in different contexts can provide insight into social-cultural norms. In doing so, this is one of the first steps in understanding how social and cultural factors in digitally mediated interactions can play a role in shaping individual attitudes and experiences of menstrual health. Nonetheless, this study is limited in its generalizability due to the purposive sampling method in the Apple App Store. Since this was not comprehensive sampling or random sampling of menstrual tracking apps across mobile platforms, these findings are not generalizable to the wider population of apps in the Google Play Store. On top of that, the word–emotion association dictionary used for the sentiment analysis was validated in English and is not yet available in Chinese, which confines our assessment to Chinese words translated to English words. Although this automated approach allowed for an efficient sentiment analysis of textual data, it was limited in taking contextual content into consideration since a word can have more than one meaning (e.g., tenderness (breast) and moderate (pain) were automatically categorized as positive even though they are negative in context). Furthermore, our study was a descriptive study with a restricted sample. Future studies should collect more data from a wider range of platforms, capture additional nuances in the range of symptoms, and test for statistically significant differences in a larger sample of English and Chinese apps. This suggests a follow-up to look at differential menstrual health outcomes for English and Chinese adolescents in other publicly available datasets. A deeper secondary data analysis is needed that takes into account varying aspects within specific countries, like

the U.S. and China, including key variables like urbanization, access to reliable technology, religion, and other relevant factors that could affect how adolescent girls interact with this technology. Additional studies should investigate the accuracy of information in menstrual tracking apps as well as decision making from user experience researchers and designers who determine default symptom features in apps. Future research can also incorporate primary data collection and an analysis of interviews, focus groups, or surveys with English and Chinese teenagers who use menstrual tracking apps to understand user needs and experiences firsthand.

## 5. Conclusions

In this mixed-methods content analysis, we explored symptoms in menstrual tracking apps in English and Chinese. The findings highlighted inadequate menstrual literacy within popular free apps, cross-cultural differences in symptoms, and a negativity bias in symptoms presented to users. Overall, the benefits and risks of using digital technology for tracking menstrual cycles need to be carefully weighed by adolescent users and their parents or caregivers. Self-managing symptoms with proper guidance and information could be beneficial in empowering girls to be aware of their own bodies and be in charge of their own menstrual cycles. For instance, commonly cited helpful reasons for using menstrual tracking apps include improving awareness of own health, taking note of bodily changes throughout the cycle, and recording unusual symptoms for conversations with healthcare providers [8]. Such digital health tools can be valuable for voluntary self-monitoring yet there are gaps in information, underscored in this study, that need to be filled in order to properly support adolescents during menstruation. In a rapidly advancing digital environment, the vast majority of teenagers have access to a smartphone. Used and implemented wisely, technology can be leveraged to improve menstrual attitudes and experiences during adolescence, elevate girls' bodily autonomy related to their sexual and reproductive health, and reduce gender inequities over time.

**Author Contributions:** Y.Y. proposed the study and collaborated with S.P. on the study design. S.P. led the English app data collection, analysis, and manuscript draft. Y.Y. led the Chinese app data collection and analysis, then assisted with the manuscript draft. M.G.H. critically revised and approved the final manuscript. All authors have read and agreed to the published version of the manuscript.

**Funding:** This research received no external funding.

**Institutional Review Board Statement:** Ethical review and approval were waived for this study because this study involved a secondary data analysis of public data.

**Informed Consent Statement:** Not applicable.

**Data Availability Statement:** The raw data used in this study are available from the corresponding author upon a reasonable request.

**Acknowledgments:** We would like to thank Kristen Fu, our undergraduate research assistant, for her contribution to this project. We would also like to thank the Haselton lab for their valuable feedback and comments. Lastly, we would like to acknowledge Yana Potashnik for her support and expertise in user experience research.

**Conflicts of Interest:** The authors declare no conflict of interest.

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
