# Peer review of "Menstrual Symptoms: Insights from Mobile Menstrual Tracking Applications for English and Chinese Teenagers"

_adolescents, doi:10.3390/adolescents3030027_

Round 1

Reviewer 1 Report

Thank you for the opportunity to review this paper. I was excited to read this manuscript because the topic of menstrual apps is of great interest to me.  Strengths: known knowledge gap, high interrater agreement 

Background: This section needs improvement. The background section does not provide an adequate argument for why this study was needed. Provide more information about why was US and China was chosen for the comparison.

Methods: Please describe mixed methods design in more detail. Was the  word-emotion association dictionary analysis validated for assessing Chinese words translated to English?  It seems like the translation could significantly change the tone.

Results: I don't understand how to interpret the finding that the most common "positive" sentiment words were breast tenderness and moderate pain. 

Conclusions: This manuscript seems to be suggesting that apps should be providing more recommendations about symptom self-management but does not provide adequate argument/sources as to why this would be beneficial/not harmful. 

Author Response

Thank you for your thoughtful commentary, we greatly appreciate your feedback and consideration. Here are our responses below:

Background Response: This comment was addressed in the attached document and the setting was elaborated on in the Introduction.

Methods Response: This comment was addressed in the attached document and revisions were made to Methods and Limitations.

Results Response: Thank you for pointing this out, this finding highlights a difficulty with automated sentiment analysis that assesses the tone of a word-unit without context, which we added to the Limitations in the attached document.

Conclusions Response: This comment was addressed in the attached document and further described in Conclusions.

Reviewer 2 Report

Dear authors:

Thanks for the opportunity to review the manuscript entitled: “Menstrual Attitudes in the U.S. and China: Insights from Mobile Menstrual Tracking Applications for Teenagers”. The article is in line with the journal’s scope. This is a fascinating and relevant topic. I have some recommendations which are outlined below. 

1.                  The date of conduction (date of sampling) should be stated in abstract.

2.                  Explain more about the setting in the INTRODUCTION.

3.                  What is the problem to be solved? What do you hope to achieve?

4.                  I am uncomfortable with this sentence: “Our study explored menstrual tracking apps for American and Chinese teenagers” because not all users of English software are American.

5.                  Please list all of the reviewed applications in one sentence.

6.          Why did you limit your search to only the App Store and not include Google Play? This represents a limitation that should be noted.

7.                  May I ask whether you have exclusively assessed menstrual applications that cater specifically to adolescent menstruation?

8.                  Have the applications you reviewed provided information to their users on managing menstruation in adolescents and using tampons or menstrual cups? Please explain. Is there apprehension among unmarried Chinese adolescent regarding the utilization of tampons and menstrual cups? Is there any literature available on this topic? For further insights into the anxiety surrounding the use of menstrual cups in Eastern countries, the article with PMID: 33714263 can be consulted.

9.                  Kindly provide the reader with additional details concerning the mixed method content analysis research method.

10.              It is advisable to present all identified categories and subcategories in a table format as it allows for a more comprehensive understanding of the findings by the reader. I feel it is not very clearly organized or integrated.

11.              In qualitative research, the determination of sample size is not typically based on a review of prior literature. Rather, it is recommended that researchers utilize theoretical or purposeful sampling techniques. Please explain more about the sampling method in mixed method content analysis. It appears that additional factors, such as the popularity of Apps, must be taken into account when determining the appropriate sample size. For instance, applications boasting an extensive user base in the hundreds of thousands may warrant special consideration.

12.              Please add the Strengths of the study.

13.              What is your suggestion for health care providers/nurse or policy makers based on your findings?

Author Response

Thank you for your thoughtful commentary, we greatly appreciate your feedback and consideration. Here are our responses below:

  1. Response: This comment was addressed in the attached document and date was added to Abstract and Methods.
  2. Response:  This comment was addressed in the attached document and the setting was elaborated on in the Introduction.
  3. Response: This comment was addressed in the attached document and further mentioned in the Introduction.
  4. Response: This comment was addressed in the attached document and the sentence was fixed.
  5. Response: This comment was addressed in the attached document and apps were added to Methods.
  6. Response: We restricted our search to the App Store due to our limited access to the iOS platform, this was noted in Limitations in the attached document .
  7. Response: Yes, we looked at menstrual applications suitable for adolescent audiences based on the age rating of the app, which was mentioned in Methods in the attached document .
  8. Response: These are interesting research questions, however we did not seek to investigate menstrual products in this study and focused on menstrual symptoms. We will look into information about and attitudes toward menstrual products, such as tampons and menstrual cups, in future research. Thank you for sharing this study, we enjoyed learning about the acceptability and safety of menstrual cups in Iranian women.
  9. Response:  This comment was addressed in the attached document and additional description was added to Methods.
  10. Response: This comment was addressed in the attached document and Table 1 and 2 were consolidated into Table 1.
  11. Response: We reviewed prior literature for guidance on total sample size and N=15, then we utilized the purposeful sampling technique for the sampling method and popularity of apps, using the total number of ratings as the metric, was one of the factors that was taken into account. We revised the Methods in the attached document to clarify.
  12. Response: This comment was addressed in the attached document and Strengths were added to the Limitations and Future Directions section.
  13. Response: This comment was addressed in the attached document and the suggestions were added to Implications.

Round 2

Reviewer 1 Report

The authors appropriately responded to comments. 

Reviewer 2 Report

I think that the authors did a great job addressing review comments. Anyhow, the authors didn’t submit a point-by-point reply to the comments.